# Development of a Forced-Choice Personality Inventory via Thurstonian Item Response Theory (TIRT)

**DOI:** 10.3390/bs14121118

**Published:** 2024-11-21

**Authors:** Ioannis Tsaousis, Amjed Al-Owidha

**Affiliations:** 1Department of Psychology, National and Kapodistrian University of Athens, 15784 Athens, Greece; 2Education & Training Evaluation Commission, Riyadh 12395, Saudi Arabia; a.owidha@etec.gov.sa

**Keywords:** Thurstonian item response theory, forced-choice items, utility judgment, ipsative scores, personality assessment

## Abstract

This study had two purposes: (1) to develop a forced-choice personality inventory to assess student personality characteristics based on the five-factor (FFM) personality model and (2) to examine its factor structure via the Thurstonian Item Response Theory (TIRT) approach based on Thurstone’s law of comparative judgment. A total of 200 items were generated to represent the five dimensions, and through Principal Axis Factoring and the composite reliability index, a final pool of 75 items was selected. These items were then organized into 25 blocks, each containing three statements (triplets) designed to balance social desirability across the blocks. The study involved two samples: the first sample of 1484 students was used to refine the item pool, and the second sample of 823 university students was used to examine the factorial structure of the forced-choice inventory. After re-coding the responses into a binary format, the data were analyzed within a standard structural equation modeling (SEM) framework. Then, the TIRT model was applied to evaluate the factorial structure of the forced-choice inventory, with the results indicating an adequate fit. Further suggestions for future research with additional studies are provided to justify the scale’s reliability (e.g., test–retest) and validity (e.g., concurrent, convergent, and divergent).

## 1. Introduction

Response bias is a significant problem in measurement theory, particularly in contexts where psychometric instruments are utilized for selection or placement (e.g., in organizations, educational institutions, and other settings). It is defined as those factors that interfere with an assessment process (e.g., within an organizational, educational, or clinical context) and cause people to give (intentionally or unintentionally) inaccurate or false answers to assessment items [1]. This phenomenon is considered one of the main issues in educational and psychological testing because it distorts the respondent’s true score [2] and, consequently, affects the validity and reliability of the test [3]. Therefore, test developers and administrators must be aware of this phenomenon and how it may affect an evaluation process to develop practices that limit its effects on test outcomes.

To describe this phenomenon, researchers often use the term response set [4]. The term set refers to a temporary reaction or the demands of a situation on a respondent (e.g., time pressure or item format), where if such reaction or demands are not present (i.e., completing the assessment tool in another time context or employing a different type of item), bias-free assessment is possible. However, some scholars argue that some response sets might reflect more stable personality characteristics (i.e., the respondent displays this attitude consistently across time and situations), and thus they prefer the term response style [5]. Today, both terms are used in the literature; some researchers prefer the term set [6,7], whereas others use the term style [1,8,9,10].

The most frequent types of response bias are as follows: (1) Faking is where people give responses that may not accurately reflect their true personal characteristics to make a more favorable impression, especially when a selection procedure is involved (fake good) [1,11]. Under other circumstances, such as the testing of persons on trial for a criminal offense, people may be motivated to present a profile of a more psychologically disturbed individual than they really are (fake bad). Both cases constitute one of the major cornerstones of personality measurement since almost all personality inventories are highly susceptible to faking [1,11]. (2) Acquiescence is where participants tend to either agree (“Yeasayers”) or disagree (“Naysayers”) to the majority of the items. The former refers to individuals who tend to concur with the majority of responses, whereas the latter refers to individuals who tend to disagree with the majority of items regardless of their content [12]. (3) Social desirability is manifested when individuals respond to an item in the way they believe most people would have responded. According to [13], almost every self-report scale (especially with personality) is confronted with the problem of social desirability. (4) Indecision (or central tendency) is where respondents prefer to choose the middle option of a rating scale, which is typically neutral or avoids expressing an opinion (e.g., don’t know, uncertain, etc.). (5) Extremity bias is where respondents tend to endorse extremes, either positively or negatively [11].

One of the most effective ways to reduce the impact of response bias (especially that of social desirability and faking) is the use of *forced-choice items* rather than rating scales, which typically make up the most frequent response format when person-based attributes (such as personality traits, attitudes, values, interests, etc.) are measured. In forced-choice items, respondents are asked to select among several equally desirable or equally less desirable statements. The statements in each item are organized in blocks with two, three, four, or even more alternative statements assessing the presented constructs. Typically, the respondents are asked to rank-order the statements or to express their preferences for the items inside each block by choosing the two statements that, based on their everyday behavior, are the “most favorable” and “least favorable”.

It has been demonstrated that using a forced-choice format effectively controls uniform response biases [14] and achieves improved operational validity coefficients (e.g., [15,16]). When compared to single-stimulus (SS) items, it has been found that forced-choice items can significantly reduce score inflation due to “faking good” [16,17,18]. It has also been found that forced-choice items help to reduce the problem of lack of differentiation among responses (i.e., the tendency to choose the same answer like “agree” or “strongly agree” across multiple items), which might cause not only serious statistical distortions due to the restriction of the range of the scores (i.e., restricted variance) but also difficulty in the interpretation of the results, particularly in selection and predictive assessments [19]. Additionally, empirical data suggest that forced-choice tests are resistant to distortion to their covariance structure and exhibit more robust psychometric properties than rating scale-based tests [16,17]. Finally, ref. [17] found that the forced-choice measure predicted a work-related criterion (i.e., counterproductive work behavior) in both conditions, whereas the conventional measure predicted the criterion variable only in the honesty condition.

Empirical evidence also suggests that forced-choice items minimize response bias by placing statements in blocks according to the degree of social desirability (all items in a block have the same level). This means that respondents can only choose one of the statements in a block. Moreover, comparing statements (often in pairs) is cognitively easier than rating items, which requires participants to assess each item separately using a variety of rating alternatives (e.g., seven or more) [20]. Interestingly, in a recently published study [21], the complexity of both methods in measuring personality traits was tested in children (5th–6th grade and 7th–8th grade) and adolescents (10th–11th grade). The results showed that the forced-choice version demonstrated a better model fit, reliability, and validity evidence than the Likert-type counterpart. Last but not least, forced-choice tests typically require less time to deliver than traditional tests because they are shorter in length and, for this reason, are particularly desirable when selection processes are involved [22].

Despite the fact that certain benefits of using forced-choice items have been highlighted, their use has also been subjected to some criticism. One of the issues that is often brought up is the scoring system [23]. In conventional single-stimulus Likert-type items, respondents are asked to rate their level of agreement using a rating scale (usually 5- or 7 points). To score these items, corresponding points are given for each response category, ranging from 1 (totally disagree) to 5 or 7 (totally agree), and persons’ scores are estimated by adding their responses to all scale items. In this approach, more points are awarded to items that respondents endorse to a higher degree. In a typical “most–least like” forced-choice item, however, all alternative statements within the block are scored. One common method to score these items is to give the “most like” statement two points, the “least like” statement zero points, and the other statements—which are all unchosen—one point each. In this case, however, each participant ends up with a total of four points per block, regardless of the statements they have chosen. As a result, all respondents get an identical final score, which prevents experts from comparing individuals who have responded to the same set of items. The only comparisons that could be made are within the responses of the same individual [24]. Such data are called ipsative data, and the corresponding scales are called ipsative scales [25].

Previous studies have also shown that ipsative data might have serious implications for the psychometric characteristics of a scale. For example, it has been shown that although two variables might be positively correlated, their corresponding correlation coefficient is always negative when ipsative scores are estimated [26]. This might affect the justification of the convergent or divergent validity of the scale. Additionally, statistics typically applied in psychometric analyses, such as regression analysis, factor analysis, and structural equation modeling (SEM), cannot be applied due to the interdependence of the variance/covariance matrices when ipsative scores are involved [27]. This might also lead to a less clear estimation and interpretation of a scale’s factor structure and construct validity [28]. Finally, ipsative scores tend to produce underestimated (sometimes overestimated) internal consistency indices (i.e., alpha) [29]. This is mainly due to the lack of variability in the scores, since all respondents end up with the same total score across items. Another reason is that ipsative data produce negative (or zero) correlations among items, thereby distorting the average item correlation and producing biased alpha estimates [30]. Finally, ipsative scores produce interdependence among the errors within a block—a factor that also produces distorted reliability estimates [31].

Apart from the impact of the forced-choice methodology on the psychometric quality of scale, several scholars argue that this methodology also has practical implications for the selection process, especially in high-stakes tests. For example, when forced-choice tests are used in contexts requiring fine differentiation of traits (e.g., leadership potential vs. teamwork skills), their accuracy may be limited unless a large number of traits are measured [32]. This may affect decision making in high-stakes environments, where precise measurements of specific qualities are crucial to selecting the most suitable candidate [33].

According to [30], the above are the main reasons why forced-choice scales are not very popular in the literature, despite their beneficial characteristics in controlling response bias. In this light, this study aims to investigate the psychometric properties of a forced-choice personality inventory designed to measure students’ personality characteristics.

## 2. Materials and Methods

### 2.1. Participants and Procedure

Two different samples were used in this study. The first sample consisted of 1484 students from the 12th grade (17 years old) and was used to select the final pool of items that would be used to build the forced-choice version of the personality test. No demographic characteristics were asked for to ensure anonymity. However, the sample was equally balanced in terms of gender. Consent was obtained from school administrators before the research assistants distributed the surveys. The second sample consisted of 1194 participants, mainly university students, and was used to examine the factorial structure of the forced-choice version of the personality test. However, 371 participants had to be removed since they provided incomplete data (they logged off at some point during the administration). The total sample used for the analysis was 823. The only demographic information that was asked for was gender and age. Regarding gender, 434 (52.73%) were males and 389 (47.27%) were females. The mean age of the participants was 19.03 (0.68).

The study was conducted in accordance with the Declaration of Helsinki and approved by the Institutional Review Board of the Education & Training Evaluation Commission (approval code: TR378-2023; date of approval: 12 January 2023). First, participants were briefly informed about the aims and the scopes of the study, the importance of their involvement, and the researchers’ contact details. Next, the participants were told that their involvement was entirely voluntary and that they could withdraw at any time. All measures used during the two phases of the study were administered electronically, and all students completed an informed consent form prior to their participation. To prevent duplicate participation, each student received a unique code to access the online platform where the tests were administered. After participants logged in, the code was no longer valid.

### 2.2. Measures

The forced-choice measure used in this study was based on the five-factor taxonomy model. This model suggests that five distinct personality dimensions can efficiently explain human behavior [34,35]. Initially, 200 items measuring different aspects of the five personality domains were developed (40 items per dimension). Typical examples of the generated items were the following: “*I prefer getting individual rather than group assignments at school*” (Extraversion); “*I’m envious of students who always accomplish their goals*” (Emotional Stability)*;* “*My teachers often say that I am very creative, and I produce original works”* (Openness); “*My classmates say that I am a tender and considerate person*” (Agreeableness); “*I am very competent in doing my homework*” (Conscientiousness).

From the 200 initially developed items, 75 items (15 per dimension) exhibiting the more robust psychometric characteristics were used to build blocks for the forced-choice version of the scale. The 75 items were arranged into separate blocks, each containing three statements (triplets), producing 25 blocks. Our goal was to develop a forced-choice test of moderate length to control cognitive complexity, fatigue, and boredom. Too many blocks increase cognitive complexity, which can overwhelm working memory. Previous research has shown that long administration times can cause systematic error due to boredom, exhaustion, or low motivation [36]. Limiting the number of blocks and corresponding options prevents participants from experiencing cognitive overload, allowing them to make more reliable decisions. Similarly, too many similar trials can result in boredom, which may cause participants to rush or disengage. A reduced number of triplets ensures that a task remains mentally stimulating while maintaining attention levels [37]. Finally, it has been argued that comparing items directly within a block may be cognitively simpler than rating them one by one, particularly when there are many rating categories with few or poor verbal anchors [20].

Before each statement entered a block, its social desirability index was estimated. The main aim was to build blocks containing balanced statements in terms of their desirability. Particularly, triads of statements with similar desirability indexes were placed into a block. By placing items with similar desirability indexes, social desirability bias could be controlled since the preference options in each block were equal [17]. Great care was also taken during the block design, where each trait (dimension) occurring only once in each triplet was compared to every other at least once and was evenly distributed across the blocks. Finally, the position of each trait within the blocks (1st, 2nd, or 3rd) was balanced across the scale. An example of an item block is presented in Figure 1.

### 2.3. Data Analysis

First, using robust psychometric techniques, each of the five generated item pools was analyzed to identify the items that best represent conceptually the five personality domains. Using exploratory factor analysis (EFA), the items with the best psychometric properties were selected to set the final item pool for each domain. Principal Axis Factoring (PAF) and Promax rotation were adopted as factor extraction and rotation strategies. The PAF approach is usually preferred during the beginning stages of the test development process since it provides a framework of analysis that is less restricted in terms of statistical assumptions (i.e., multivariate normality) and because it helps to identify factors that are less robust but might be important in terms of content validity [38]. The Promax method of rotation is preferred to the widely applied Direct Oblimin method due to its conceptual and computational simplicity, although, in practice, they produce the same results [39]. To decide the number of meaningful factors that could be retained after the factors’ rotation, Parallel Analysis (PA) [40] was used.

During the item analysis, the sequential item selection approach was applied, where EFA was used in a stepwise fashion [41,42]. With this approach, each scale’s conceptual cohesiveness was maximized by identifying and removing, one by one, those items that either failed to load significantly on a main factor (i.e., factor loading < 0.30) or cross-loaded on more than one factor (i.e., factor loading > 0.40) or did not share a large enough common variance with the rest of the domain’s item (communality < 0.50). At the end of this step, we ended up with a factor structure containing only items that best described the conceptual model (i.e., a five-factor model).

The composite reliability (CR) index or omega [43,44] was estimated to examine the internal consistency of each factor scale. The use of the CR index over the well-known alpha index was preferred since it appears to be a more robust method for examining internal consistency [45,46]. Several scholars have recently argued that the alpha index should be used only when test items evince tau-equivalence [47,48]. However, such a condition rarely occurs in the real world since it presumes equal loadings from all manifest variables [49]. On the other hand, the omega index seems more appropriate since it has been designed for scales where items vary in how strongly they are related to the construct being measured (factor loading equality is not assumed). Moreover, omega is a suitable scale index when all items contribute equally to the scale’s total score (unit-weighted scales) [50,51].

Finally, to examine the factor structure of the forced-choice version of the test, ref. [30] provided a new approach to overcome the issue with the ipsative nature of the scores. The authors suggested a methodology for scoring forced-choice responses within the context of an SEM framework, basing it on the law of comparative judgment [52]. When an individual is asked to compare various statements following this law, they examine each statement separately and pick the response that corresponds more closely to their behavior, referred to as “utility judgment”. Then, they select the statement with the largest utility among the available options. In practice, a utility index is generated for each item, and a person *p* will rank an item *i* higher than an item *k* if the utility index that they generate for item *i* is greater than the utility index that they generate for item *k*. The following formula can be used to explain this procedure mathematically:(1)$$\mathrm{Preferred}\;\mathrm{item}=\left\{\begin{array}{c} i\;if\;{utility}_{pi}\geq {utility}_{pk}\\ k\;if\;{utility}_{pi}<{utility}_{pk} \end{array}\right.$$

Following this conceptual framework, Ref. [29] introduced the Thurstonian Item Response Theory (TIRT) statistical method to calculate a variety of psychometric properties, such as item parameters, correlations among latent factors, and person estimates across multiple dimensions within the context of SEM. The utility of each statement is an unobserved (latent) variable that underlies each item, to use the terminology of latent variable modeling. The observed variable represents the result of each implicit comparison (i.e., whether the utility of statement A is better than those of statements B and C) within each block. As a result, the utilities of items are modeled as latent variables that are assumed to determine the observed outcomes.

To analyze responses involving forced choices utilizing the Thurstonian IRT model, all possible comparisons between items within each block are evaluated by applying the utility rule described in (1). In the case of a triplet block (A, B, and C), for instance, three paired comparisons are carried out: {A, B}, {A, C}, and {B, C}. A binary variable known as *yl* is used to denote the result of each comparison for each pairwise combination represented by the formula *l* = {*i*,*k*}:(2)$$yl=\left\{\begin{array}{c} 1\;if\;item\;i\;is\;prefered\;to\;item\;k\\ 0\;if\;item\;k\;is\;prefered\;to\;item\;i \end{array}\right.$$

The example presented in Table 1 shows how the selected options within a block are transformed into binary outcomes. In this scenario, the respondent is prompted to choose the “most favorable” and “least favorable” statements from among the three choices (A, B, and C). Although many different alternative combinations of selected options are possible, let us hypothesize that the respondent selects option A as “most favorable” and option B as “least favorable”, leaving option C unselected. Using Formula (2), it is possible to achieve the following three binary outcomes:

The pair {A, B} receives a score of 1 because attribute A is more favorable than attribute B (the person indicated option B as least favorable), and the pair {A, C} receives a score of 1 because attribute A is more favorable than attribute C (the person did not indicate option C as most or least favorable, so it is scored 0). The pair {B, C} receives a score of 0 because attribute B is the least favorable attribute among the three options, and therefore attribute C is more favorable than B. These binary outcomes, which are the products of a person’s utility from all possible comparative judgments of the statements, are assumed to be related to the measured dimensions (e.g., the Big Five dimensions) using a linear factor analysis model known as the Thurstonian factor model [30]. Interestingly, a Thurstonian IRT model is the same as a second-order factor analysis model when binary data are used [20]. Therefore, each paired comparison (i.e., the result of each implicit comparison) can be described as the linear combination of a personality trait using the following formula:(3)$${utility}_{pi}={mean}_{i}+{loading}_{i}\;{trait}_{p}^{a}+{error}_{pi}$$

As a result, to calculate a utility index, which is the probability *P*(*y_p_*_{*i*,*k*}_ = 1) of a person *p* selecting item *i* over item *k* in a pair of statements {*i*,*k*} inside a block assessing various personality traits *a* and *b*, the following formula could be used:(4)$$P(y_{p\{i,k\}}=1)=\Phi (\frac{-{threshold}_{\left\{i,k\right\}}+{loading}_{i}\;{trait}_{p}^{a}-{loading}_{k}\;{trait}_{p}^{b}}{\sqrt{var({error}_{i})+var({error)}_{k}}})$$
where *Φ* (x) denotes the cumulative standard normal distribution function evaluated at x.

Interestingly, for items with positive factor loadings, the likelihood of selecting statement *i* over statement *k* increases as the score on the dimension measured by statement *i* increases. In contrast, the score on the dimension measured by statement *k* decreases. Additionally, it was demonstrated that Thurstonian IRT models could be simply implemented within an SEM framework [20]. It is, therefore, simple to estimate and evaluate these models using well-known statistical tools (e.g., Mplus version 8.5). As mentioned earlier, this model is mathematically equivalent to its Thurstonian IRT (TIRT) counterpart [20].

However, despite the promises that the TIRT model brought to the forced-choice methodology, there are scholars who argue that the practical utility of this approach is not so evident [53]. For example, it has been suggested that the precision of scale and person estimates when the TIRT approach is implemented depends on several test characteristics (e.g., an equal number of positively and negatively keyed items within blocks should be ensured) which sometimes may not be easily addressed [31]. Recent findings have also suggested that how items are put into the blocks (e.g., whether dominance items are included) could also affect the item and person parameter estimation [54,55].

Finally, to evaluate how well the model fits the data, several different fit indices, including the chi-square (χ^2^) statistic and related degrees of freedom (*df*), the Comparative Fit Index (*CFI*), the Tucker–Lewis Index (*TLI*), the Root Mean Square Error of Approximation (*RMSEA*), and the Standardized Root Mean Square Residual *(SRMR*) were estimated. Guidelines for what constitutes a good fit can vary; however, a CFI and a TLI greater than 0.95 (with values >0.90 demonstrating reasonable fit), an RMSEA equal to or less than 0.06 (with values of 0.08 demonstrating reasonable fit), and an SRMR equal to or less than 0.08 are thought to represent a good fit [56,57].

## 3. Results

### 3.1. Preliminary Data Analysis

Before the main analysis, data screening techniques were conducted to ensure that all necessary statistical assumptions were met. Particularly, univariate statistics (i.e., means, standard deviations, skewness, and kurtosis) were examined to identify items exhibiting deviant characteristics. For example, items with a mean value of <1.5 or >3.5 were removed from the item analysis due to low variability in persons’ scores. Similarly, items with skewness and kurtosis values of >1.96 were removed due to non-normally distributed scores. Twelve items were removed from the subsequent stages of the analysis.

### 3.2. Item Analysis and Final Item Pool

The second step involved selecting the items to construct the forced-choice version of the test. Particularly, using factor analysis (i.e., PAF), the items with the best psychometric properties were selected to set the final item pool for each domain. To decide the number of meaningful factors that should be retained in each RAF analysis, a Parallel Analysis was run using the Monte Carlo PCA for Parallel Analysis software (version 2.3) [58]. The results from Parallel Analysis showed that the one-factor model was the optimum model for every domain. Next, following a stepwise approach and after several consecutive factor analyses, 15 items with the highest factor loadings in each domain (15 × 5 domains = 75 items) were selected to set the final item pool.

The percentage of variance explained by the selected items in each factor was 46.8%, 37.67%, 43.28%, 49.70%, and 43.25% for Extraversion, Emotional Stability, Openness, Agreeableness, and Conscientiousness, respectively. The composite reliability of the omega index was used to examine the extent to which all items in each domain measure the same underlying construct. The results from the analysis showed that all items in each domain were very homogeneous: Extraversion = 0.811, Emotional Stability = 0.769, Openness = 0.823, Agreeableness = 0.877, and Conscientiousness = 0.814. The selected items and their factor loadings for each domain are presented in the Appendix A. SPSS package 26.0 was utilized to conduct the exploratory factor analysis and the reliability analysis.

### 3.3. Examining the Factor Structure of the Forced-Choice Version of the Test via T-IRT

Before fitting the T-IRT model in Mplus, the participants’ responses were coded as binary outcomes using an Excel macro [28]. This Excel macro was also utilized to construct a corresponding Mplus syntax. The model investigated was a five-factor model corresponding to the Big Five theoretical framework [34]. To ensure the model converged effectively, factor intercorrelations were fixed [23,28,59] using data from a prior study, where the five-factor model of personality was examined in the same cultural context [60]. This model is shown in Figure 2.

Due to the binary nature of the data, the DWLS algorithm was used, and the model was fitted to tetrachoric correlations. The hypothesized structural model was that of independent clusters [37] with no cross-loadings. The fit for the hypothesized five-factor model was as follows: χ^2^ = 7022.76, *df* = 2650; CFI = 0.869; TLI = 0.862; RMSEA = 0.045 (90% CI = [0.044, 0.046], and SRMR = 0.078. As can be seen, both the RMSEA and the SRMR absolute fit indices suggested an adequate fit to the data, while the other two incremental indices (CFI and TLI) indicated a close-to-adequate fit. However, it is common for the goodness-of-fit indices like CFI or TLI to provide lower values when using the Thurstonian IRT approach. These models are inherently complex because they involve the estimation of thresholds and item discriminations, often leading to a large number of parameters. This complexity can result in a lower fit if the model is not well-specified or if the data do not perfectly match the model’s assumptions (see [59,61]).

## 4. Discussion

Due to their effectiveness in decreasing or at least controlling response bias (e.g., faking, social desirability, etc.), forced-choice measurement models have recently been (re)utilized in the field of educational and vocational evaluation (especially if selection methods are involved). This is mainly due to the introduction of a novel statistical approach known as Thurstonian Item Response Theory (TIRT), which enables researchers to analyze ipsative scores (which are generated when forced-choice items are used) using traditional psychometric approaches such as CFA and IRT.

In this study, the factorial structure of a forced-choice version of a personality inventory based on the theoretical framework of the five-factor model of personality [34,35] was examined. Due to the ipsative nature of the produced scores, the Thurstonian IRT approach was preferred. The forced-choice version of the personality inventory consists of 25 blocks, each containing three statements for different personality traits (75 items in total). Next, we coded the forced-choice responses in each block following the scoring procedure described earlier (see the Methods section, Data Analysis). For each block, three pairwise comparisons were attempted, and three new binary (0, 1) variables were generated.

The results from the TIRT analysis provided evidence for adequate fit, although not all goodness-of-fit indices supported this finding. However, this outcome is consistent with previous research indicating the inherent complexity of Thurstonian models, which often result in lower fit indices due to the intricate estimation of parameters [59,61].

Given this, the results suggest that the forced-choice version of the new personality test is an adequate tool for measuring student personality characteristics. This finding has both theoretical and practical implications for personality assessment, especially in the context of selection and evaluation of personality characteristics. First, it adds to the growing body of literature on the effectiveness of forced-choice tests in reducing response bias in self-report measures [14,16,17,18].

Second, it demonstrates that the Thurstonian Item Response Theory (TIRT) is a robust statistical approach for analyzing ipsative data and making possible participant comparisons [23,30]. Third, this study’s findings strengthen the theoretical foundation of the five-factor model (FFM) of personality, validating its applicability within a forced-choice format. This supports the universality and flexibility of the FFM as a framework for understanding personality across various measurement formats.

From a practical standpoint, this study offers a psychometrically robust tool since it provides bias-resistant results, which, in turn, leads to more accurate evaluations of students’ personality characteristics. By reducing the impact of social desirability bias, teachers and counselors can better understand students’ true characteristics, leading to more effective and tailored educational guidance.

There are, however, several points to highlight. First, although the findings from this study provided convincing evidence regarding the scale’s construct validity, since the hypothesized five-factor model seems to fit the data adequately, further validation will strengthen the psychometric robustness of the scale. Particularly, the forced-choice version of the test should be administered along with scales measuring various aspects of human behavior (e.g., self-esteem, anxiety, happiness, empathy, etc.) to validate the scale’s concurrent, convergent, and divergent validity.

Second, the reliability of the five scales of the inventory should also be examined. However, previous studies have shown that reliability indices, such as alpha and omega, tend to produce distorted and unreliable estimates when ipsative scores are obtained. This occurs because, in ipsative data, the inter-item correlations are not produced from items that are independent of each other, as assumed in standard indices. In ipsative data, choices are forced and dependent on each other, meaning the sum or total score is constant across items [29]. To overcome this problem, several scholars have argued that test–retest reliability might be a more appropriate alternative for justifying the reliability of a forced-choice test [27,62]. For that, future research should consider conducting a study in which a sample of participants complete the scale twice, within an interval of approximately four weeks between the two administrations. Additionally, since no demographic data were provided in this study, it might be interesting for a future study to examine how trait scores from a forced-choice scale vary across variables such as gender, age, and education level.

Finally, an interesting issue arose from an anonymous reviewer regarding the effect that ipsative items may have on scoring when items that are not scored within a block (items that have not been chosen by the participant) are additionally post-coded by researchers. This coding procedure for ipsative items implies that the non-rated personality items are post-scored with a higher value (i.e., 1 point) than the active “least like” answer (i.e., 0) given as a self-report answer by participants. This opens the question of whether Thurstonian law could be diagnostically fairly applied to ipsative items because at least one alternative (namely, C) is an other-rating and not a self-rating of a participant. It might be worth examining in a future study whether the coding of ipsative items could or should be reduced to the dichotomy (1 = most like vs. 0 = least like) that always refers to the self-chosen answers of the participants while leaving the non-chosen item alternative aside for data analysis.

## 5. Conclusions

To sum up, this study aimed to examine the factorial structure of the forced-choice version of a new test measuring student personality via the Thurstonian Item Response Theory (TIRT) approach. The findings revealed an acceptable model fit, demonstrating the potential of the forced-choice format combined with the TIRT model to reduce response biases in personality assessment. The findings support the continued use and development of such tools, particularly in educational settings, where accurate measurement of personality traits is important for selection, placement, and other evaluative purposes.

## Figures and Tables

**Figure 1 behavsci-14-01118-f001:**
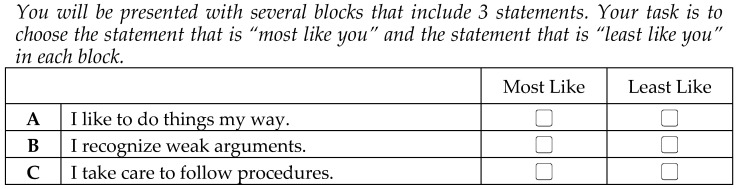
Example of a forced-choice item (triplet).

**Figure 2 behavsci-14-01118-f002:**
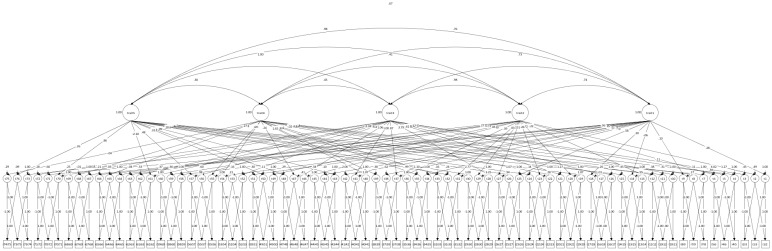
The personality scale measurement model under the Thurstonian Item Response Theory approach. Latent item utilities are denoted t1–t75; observed dummy contrasts are denoted according to the items contrasted; for example, {i1i2} represents the contrast between items 1 and 2.

**Table 1 behavsci-14-01118-t001:** Example of how a forced-choice item is scored within the Thurstonian framework.

			Binary Comparisons		
A	B	C	{A, B}	{A, C}	{B, C}
Most	Least		1	1	0

## Data Availability

The data that support the findings of this study are available from the Education & Training Evaluation Commission (ETEC). Restrictions apply to the availability of these data, which were used under license for this study. Data are available from the authors upon reasonable request and with the permission of the ETEC.

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
