# Peer review of "Development of a Forced-Choice Personality Inventory via Thurstonian Item Response Theory (TIRT)"

_behavsci, 2024, doi:10.3390/bs14121118_

Round 1

Reviewer 1 Report

Comments and Suggestions for Authors

Review of the manuscript: behavsci-3199525

The present manuscript investigates the five-factor model of personality by means of ipsative items. I carefully read the manuscript. Subsequently, I summarize my questions and recommendations:

Introduction

The Introduction is widely well written. References have been cited for most of the arguments.

The argumentation in lines 87 to 89 could be extended: “Second, forced-choice items help … This phenomenon is also known as the “halo effect” [19]. Reference to the “halo effect” has been argued in a skippy way. Please outline why a “halo effect” could take effect when participants answer rating scales resulting in an advantage for using forced-choice items.

Lines 93 to 95: The authors state: “Moreover, comparing statements (often in pairs) is easier cognitively than rating items, which requires participants to assess each item separately using a variety of rating alternatives (e.g., seven or more) [20]. ” Could the authors cite studies that have investigated the self-reported cognitive demand when participants have answered items with rating scales and ipsative items, respectively?

Lines 110 to 112: The authors describe the scoring method of answered ipsative items. I recommend the authors address at least in the Discussion the effect that ipsative items provide the special situation that the items that are not scored as “most like” (score: 2) and “least like” (score: 0) are additionally post-coded by researchers. This coding procedure for ipsative items implies that the non-rated personality items are post-scored with a higher value -namely 1- than the active “least like” answer that is given as a self-report answer by participants. The question arises of whether the coding of ipsative items could or should be reduced to the dichotomy (1 = most like, 0 = least like) that always refers to the self-chosen answers of the participants while leaving the non-chosen item alternative aside for data analysis.

Lines 122 to 125: Is there a more representative set of published studies that investigated the effects of the issue “…internal consistency scores in ipsative scales are heavily distorted … since true and error scores are affected across scales [28].”? Could the authors explain in an extended way how distortions on reliability scores are related to “affected true and error scores across scales”?

Materials and Methods

The authors refer to impressively large samples. For sample 1, no demographic variables have been stored. Due to missing demographic information I wonder: How could the author control for double participation?

Please also report whether the examination took place digitally or in presence. Please report the instruction that obtained participants prior to answering the items. Did each participant in samples 1 and 2 provide written informed consent prior to the examination? Please report this detail explicitly in the text. Did the authors have an approval of an Ethics Committee for examinations with both samples?

Lines 165 to 166: Please provide an example of an item block.

The authors refer to a parallel analysis to decide on the number of factors that could have been rotated. Please provide a Figure illustrating the empirical and random eigenvalues. Which software did the authors use to perform the random eigenvalues? In addition, please also report the software that has been used for SEM. Line 265: “(e.g., Mplus)” has been mentioned. However, it is not clear whether Mplus has in fact been applied in this study and for which version.

Line 188: Why do the authors refer to a “composite reliability (CR) index? Please provide a reference and please describe the difference between a composite reliability and reliability scores as mostly reported in psychometric studies.

Line 206 to 207: Please describe the diagnostic purpose of the utility index.

With regard to the utility index: Could the authors explain the calculation for an item block of their items and the decision that results from the utility index for their item selection?

Line 228, Table: Please provide a Table caption.

Line 240: The authors write: “…and, therefore, attribute C is more favorable than B.”. However, C refers the post-scored items. Thus, the description that C would be a more favorable alternative is an interpretation of the examiner and no active response of a participant. This opens the question of whether Thurstonian law could be diagnostically fairly applied to ipsative items because at least one alternative (namely: C) is an other-rating and no self-rating of a participant.

Moreover, the suggested alternatives that result in binary comparisons are reduced for alternative C to one option that, however, comprises three different -non-chosen- alternatives. Thus, the question is: Might participants be thought of at least seven instead of three choices? – Namely: A, B / A, C1 / B, C1 / A, C2 / B, C2 / A, C3 / B, C3?

Should the Table be extended to illustrate participants’ choices more completely?

Line 266: “As mentioned earlier, this model is mathematically equivalent to its Thurstonian IRT (TIRT) counterpart [29].” The cited reference does not provide the mathematical evidence. Please describe which results or data in Brown et al. (2011) are the basis for the summary of the authors (“..is mathematically equivalent..”).

Line 274 to 275: “It is important to point out that a Thurstonian item response model employs tetrachoric correlations of dummy-coded binary outcomes.”. The connection between a special type of correlation analysis (tetrachoric correlations) and dummy-coded binary outcomes is not clear. The aim the sentence for the manuscript is not clear. The sentence is no summary but opens a new idea that requires more reference and description in relation to the aims of the study. Alternatively, the authors could think of deleting the sentence as the Results section does not seem to provide results that refer to this sentence (tetrachoric correlations and dummy-coded binary outcomes).

Results

Line 291: What were the criteria for selecting 15 items per domain?

Tables S1 to S5 provide the factor loadings. Without transparency for the results of the parallel analysis readers could not independently evaluate whether five single factors or five factors at once should have been extracted. How could the authors conclude based on one parallel analysis that five single domain factors should be extracted which do not allow for a factor rotation?

Line 311: According to the reported result of the parallel analysis there should be five single domain factors. The authors report one series of model fit indices. Based on the authors’ description I would have expected five series of fit indices – namely one series of fit indices per domain. Please might the authors explain?

Line 316: Please provide a reference for the sentence: “However, it is common for the goodness of fit indices like CFI and TLI to provide lower values when using the Thurstonian IRT approach.”

Discussion

Line 365: The authors investigated the factorial validity of the ipsative items for the five-factor model. Thus, the empirical evidence of their study rather refers to construct validity than to content validity because five factors have been reported to emerge. If there would be even a larger number of personality factors -as suggested in cross-cultural studies for the five-factor model- has not been investigated in this study. Thus, the content for the five-factor model could be different to the tested construct validity.

When I had read the complete manuscript, I wondered whether the authors have reported the results for sample 2 exclusively. Could sample 1 be left aside for data analysis because no demographics have been explored? Would this revision (i.e., referring to the sample with demographics) alter the description in the Discussion (line 380)?

Reviewer 2 Report

Comments and Suggestions for Authors

The study described in this manuscript focused on the development of a forced-choice personality inventory to assess student personality characteristics based on a five-factor personality model and then examined its factor structure via the recently popularized so-called Thurstonian Item Response Theory (TIRT) approach. Forced-choice items essentially involve compulsory responses from studied participants on several equally desirable measurement instrument statements. The results indicated that an acceptable overall model could be fit to the data and supported the potential use of the forced-choice format combined with the TIRT model in personality assessment research.

Overall, this is an excellent manuscript that will be of great interest to applied researchers involved in the study of personality characteristics and behaviors. Perhaps even by just adding a simple note in the figure will suffice to address my noted concern. 

The one issue that perhaps needs some clarification is the currently listed value for the degrees of freedom of the tested model (stated to be df = 2650). The specified degrees of freedom presumably correspond to the characteristics of the tested model presented in accordance with Figure 1. However, if as it is also stated on page 7 (line 306), the factor intercorrelations were fixed, then the provided degrees of freedom seem unclear to me. There are clearly 75(76)/2 = 2850 free parameters in the proposed covariance data structure. Obviously 75 + 75 = 150 parameters correspond to the item loadings and error variances. So where are the remaining 50 parameters constraints coming from? There are no factor intercorrelations, so it is not evident where the rest are coming from to reach df = 2650.

Reviewer 3 Report

Comments and Suggestions for Authors

I would like to express my appreciation to the authors for presenting such a solid and well-structured manuscript. The study demonstrates a thorough application of the Thurstonian Item Response Theory and offers valuable insights into the development of a forced-choice personality inventory. The clarity of the results is commendable. To further enhance the manuscript's contribution to the field, I would suggest integrating more recent research, deepening the theoretical discussions, and slightly expanding the reference list. These improvements would undoubtedly strengthen the scholarly impact of this excellent work.

  1. Is the content succinctly described and contextualized with respect to previous and present theoretical background and empirical research (if applicable) on the topic?
    Can be improved
    The manuscript provides a good overview, but some sections could better integrate recent theoretical advancements and empirical research to enhance context.

  2. Are the research design, questions, hypotheses and methods clearly stated?
    Yes
    The research design, questions, hypotheses, and methods are clearly presented and well-structured.

  3. Are the arguments and discussion of findings coherent, balanced and compelling?
    Can be improved
    While the arguments are generally coherent, there are areas where a more balanced discussion and stronger connections between findings and theoretical implications could strengthen the overall argument.

  4. For empirical research, are the results clearly presented?
    Yes
    The results are clearly presented and well-organized.

  5. Is the article adequately referenced?
    Yes

  6. Are the conclusions thoroughly supported by the results presented in the article or referenced in secondary literature?
    Can be improved
    The conclusions are generally supported by the results, but further integration of secondary literature could provide additional support and depth.

  1. Originality
    Average
    The manuscript presents relevant research, but it does not significantly advance new ideas or concepts in the field.

  2. Contribution to Scholarship
    Average
    The contribution to the field is solid but could be enhanced by addressing gaps or providing a more novel perspective.

  3. Quality of Structure and Clarity
    High
    The manuscript is well-structured and clear, with a logical flow of information.

  4. Logical Coherence/Strength of Argument/Academic Soundness
    Average
    The arguments are generally sound, but some areas lack depth and coherence in connecting findings to broader theoretical discussions.

  5. Engagement with sources as well as recent scholarship
    Average
    The engagement with recent scholarship is limited. Including more up-to-date sources and discussions would strengthen the manuscript.

  6. Overall Merit
    Average
    The manuscript has merit but requires improvements in several areas to meet higher standards of academic research.

  7. Are the references cited in this manuscript appropriate and relevant to this research?
    High
    While some references are appropriate, the inclusion of more recent and relevant studies would enhance the manuscript.
